# Adaptation to hydrostatic pressure modulates proteome dynamics in corrosive sulfate-reducing bacteria

Nicolò Ivanovich,[1] Xue Guo,[2] Radoslaw M. Sobota,[2] Federico M. Lauro[1,3]

ABSTRACT  Microbially influenced corrosion (MIC) poses a significant threat to metal structures across various industrial sectors, leading to substantial economic losses and potential environmental damages. As deep sea exploration and infrastructure development expand, understanding MIC under high hydrostatic pressure becomes increasingly critical. Microorganisms in these extreme environments must undergo specific structural and metabolic adaptations to survive and thrive. In this study, we employed a proteomic approach to investigate the physiological states and corrosive potential of two sulfate-reducing bacteria (SRB) with different hydrostatic pressure optima. By simulating depths ranging from the sea surface to 3,000 m, we identified species-specific corrosion mechanisms and distinct proteomic profiles associated with pressure adaptation. These findings reveal opposing trends in corrosion rates, emphasizing the complex relationship between microbial physiology and environmental conditions. This study underscores the importance of characterizing microbial responses to hydrostatic pressure for improving corrosion risk assessments and predictive models, particularly for metallic structures deployed in extreme environments.

IMPORTANCE  Microbially influenced corrosion (MIC) is a widely studied phenomenon that continues to be poorly understood due to its inherent multifaceted nature. MIC involves complex interactions between microbial communities, their metabolic activities, and the surrounding environmental and material conditions. With the recent rapid expansion of human exploration into previously inaccessible areas, such as the deep sea, new questions about how the physiological adaptations of microbial communities influence their corrosive capabilities have been raised. This study investigates the relationship between corrosion in sulfate-reducing bacteria and the proteomic responses to environmental stresses in differently adapted organisms. It suggests that microorganisms sharing the same core metabolic pathways can drive corrosion through different mechanisms and highlights how hydrostatic pressure adaptations can impact MIC severity.

KEYWORDS  proteomics, high hydrostatic pressure, sulfate-reducing bacteria, microbially influenced corrosion

Editor Katharina Kujala, Luonnonvarakeskus Soil Ecosystems, Oulu, Finland

Peer Reviewer Satoshi Wakai, Japan Agency for Marine-Earth Science and Technology (JAMSTEC), Yokosuka, Kanagawa, Japan

Address correspondence to Nicolò Ivanovich, nicolo.ivanovich@lnu.se, Federico M. Lauro, federicolauro@hotmail.com, or Radoslaw M. Sobota, Radoslaw_Sobota@a-star.edu.sg.

Nicolò Ivanovich and Xue Guo contributed equally to this article. Authors order was determined based on who conceived the study.

The authors declare no conflict of interest.

See the funding table on p. 17.

Microbial influenced corrosion (MIC) is one of the most significant challenges across all industries where metal equipment is employed (1, 2). Every year, MIC-related failures cause a global expense of billions of US dollars and cause considerable environmental impacts (3). The constant expansion of human activities in remote areas, characterized by extreme conditions, has the potential to further increase the risk of MIC-induced failures.

The deep sea is among the most intensively exploited extreme environments (4). It is characterized by high hydrostatic pressure (HHP), darkness, limited nutrients availability, and low temperatures, with the exception of hydrothermal vent regions (5–7). Despite

these seemingly harsh conditions, the deep sea hosts a diverse array of microorganisms known as piezophiles (8, 9) and represents the largest ecosystem on Earth by volume (10). Numerous studies have investigated adaptations to life under HHP conditions, identifying many mechanisms that overlap with general stress responses and others that are uniquely specialized for HHP (11–16). However, although there is an increasing interest in the abiotic corrosion processes and feasible mitigation strategies under HHP conditions (17), the effects of metabolic adaptations to the deep sea on bacterial corrosive capabilities, as well as the role of environmental biomarkers in the prediction and mitigation of MIC, remain poorly understood.

Microorganisms endemic to the deep sea habitats can be found as free cells or in communities forming biofilms on virtually any available surfaces, metal included (18–21). The formation of a biofilm on the metallic surfaces plays a crucial role in microbial corrosion processes (22–24) with extracellular polymeric substance (EPS), which represents 90% of the total volume of a biofilm, guaranteeing additionally protection from external stresses (25–27).

Under aerobic conditions, microorganisms within a biofilm can indirectly promote corrosion with the generation of areas with varying oxygen concentrations on the surface of the metal (28, 29). Additionally, the development of anaerobic regions provides niches for the growth of other corrosive species (30–32). It is under anaerobic conditions that MIC overtakes abiotic corrosion as the primary cause of material degradation. This process involves various microbial species, each contributing through distinct mechanisms (28, 29, 33, 34), with one notable group being sulfate-reducing bacteria (SRB). SRB respire anaerobically, using sulfate as an electron acceptor (35, 36), and contribute to MIC through multiple mechanisms. They promote MIC either directly by consuming electrons released during the natural dissolution of metals (37, 38) or indirectly through the production of corrosive metabolic by-products, such as hydrogen sulfide ($H_2S$) (39, 40).

SRB can be found in many habitats, including the deep sea. Interestingly, several SRB species have been isolated from depths greater than 200 m (41–44), which is commonly considered as the initial depth of the deep sea, but only a few studies have analyzed their proteomic profiles and corrosive capabilities (21, 45–48).

This study aimed to uncover the impact of HHP on the metabolism of SRB and their corrosive potential. LC-MS/MS-based untargeted proteomics was used to investigate changes in large-scale protein profiles of two SRB species under varying hydrostatic pressure conditions. *Pseudodesulfovibrio profundus* 500-1, a piezophilic bacterium isolated from deep sediment samples in Japan Sea (42), and *Desulfovibrio ferrophilus* IS5, a non-piezophilic bacterium commonly employed in MIC studies (49–51), were selected as pressure-adapted and non-pressure-adapted model organisms, respectively.

The experiments were conducted at three simulated depths (0, 1,500, and 3,000 m) to represent shallow waters, the optimal HHP conditions for the piezophilic species, and the deepest depth achievable without compromising the viability of the non-piezophilic control. Both species were cultured individually in oligotrophic artificial seawater (ASW) in the presence of marine-grade steel to explore the proteomic responses in planktonic and biofilm-forming cells, identifying species-specific responses and distinct adaptations to hydrostatic pressure. Furthermore, the results suggest that corrosive capabilities of the two bacteria were driven by different mechanisms and closely related to their optimal pressure adaptation. A positive correlation between corrosion rate and HHP was observed in the pressure-adapted species, while the opposite trend was observed for the non-adapted organism.

## RESULTS

### Hydrostatic pressure induces variations in corrosion rates

To evaluate the impact of pressure on the corrosion capabilities of the piezophilic *P. profundus* and the control strain *D. ferrophilus*, corrosion rates were derived from weight loss measurements of coupons incubated at 0.1, 15, and 30 MPa. (Fig. 1).

*P. profundus* exhibited a positive correlation between corrosion activity and hydrostatic pressure, with an initial rate of 0.005 ± 0.0002 mm/year at 0.1 MPa, rising to 0.0058 ± 0.0006 mm/year at 15 MPa, and reaching a maximum value of 0.0079 ± 0.0004 mm/year at the 30 MPa (Fig. 1).

In contrast, at 0.1 MPa, the corrosion rate of *D. ferrophilus* was determined to be 0.106 ± 0.012 mm/year. As pressure increased to 15 MPa, the rate only slightly decreased to 0.0852 ± 0.004 mm/year. However, a marked decline was observed at the highest pressure (30 MPa), where the corrosion rate dropped to 0.008 ± 0.001 mm/year.

These opposite trends highlight the distinct corrosion capabilities of the two bacterial species, likely driven by differences in metabolic states and biofilm formation.

### Proteome analysis reveals distinct patterns driven by hydrostatic pressure and physiological state

The proteomic differences between biofilm and planktonic cells across the two bacterial species were initially examined in response to increasing pressure. A data-dependent LC-MS/MS-based proteomic approach was employed to analyze both biofilm and planktonic proteomes under three different pressure conditions. Characterization of both biofilm and planktonic cells by shotgun proteomics led to the identification of a total of 3,168 proteins in *P. profundus* and 2,842 proteins in *D. ferrophilus* cultures, respectively.

Principal component analysis (PCA) was performed to visualize similarities and differences across different experimental conditions. Both species exhibited markedly

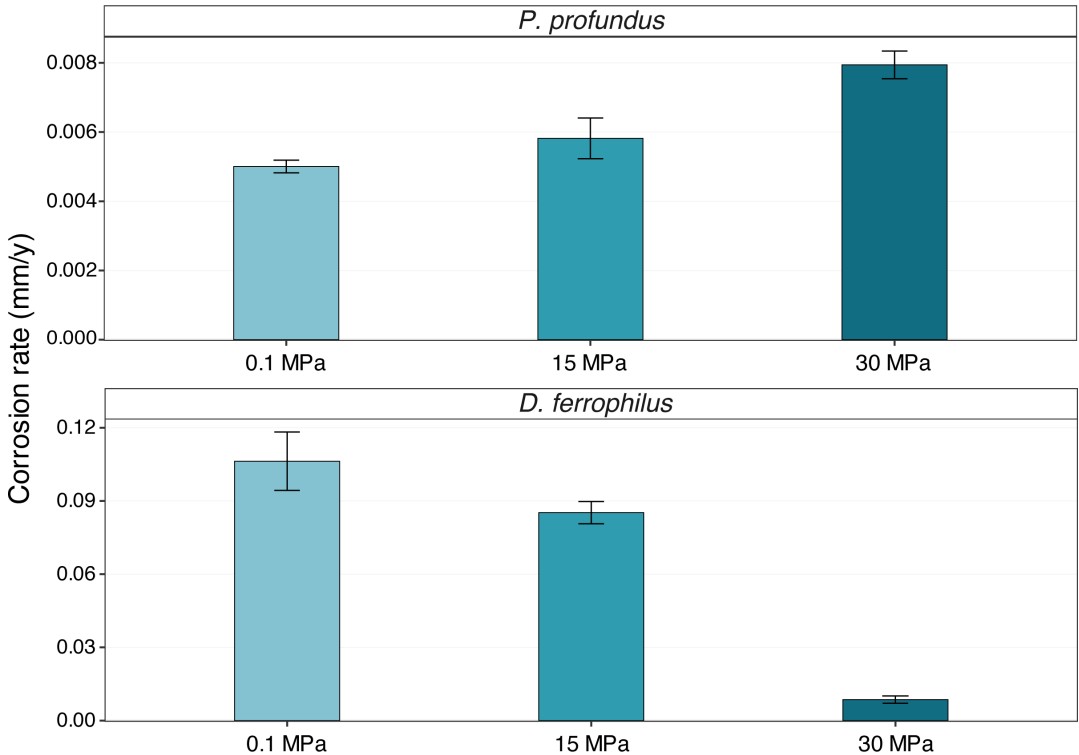

**FIG 1** Corrosion rates of AH36 coupons increase with increasing hydrostatic pressure in the presence of *P. profundus*, while they decrease in *D. ferrophilus* cultures. Error bars indicate the calculated standard deviations of three biological replicates.

different proteomic signatures in their biofilm and planktonic forms (Fig. 2A and D). Subsequent in-depth analyses were carried out to assess how hydrostatic pressure influences the physiology of both biofilm and planktonic cells across the two species. In *P. profundus* cultures, a pronounced separation between proteins from biofilm and planktonic cells was observed, with the PC1 axis explaining about 51% of the total variance (Fig. 2A). However, comparisons of biofilm and planktonic proteomes across the different pressure conditions displayed a relatively homogeneous distribution, particularly within the biofilm-associated cells. (Fig. 2B and C). *D. ferrophilus* cultures also showed distinct separation between biofilm and planktonic cells, as reflected by the contribution of the PC1 axis (27.5%) to the total variance (Fig. 2D). In this case, protein expression was more responsive to hydrostatic pressure changes, with a cluster observed for planktonic cells at 30 MPa in the overall comparison (Fig. 2D), as well as pressure-specific clustering for both biofilm and planktonic samples (Fig. 2E and F).

## Unique proteomic signatures reflect distinct corrosion mechanisms and adaptive strategies.

In *P. profundus* cultures, proteome screenings revealed distinct protein expression patterns at 0.1, 15, and 30 MPa. Comparison of the biofilm proteome to planktonic cell yielded a total of 755, 680, and 687 proteins showing significant differential expression at these pressures, respectively. Volcano plots indicated that most differentially expressed proteins had log2 below 0, suggesting that overexpressed proteins in planktonic cells were significantly more abundant than those in biofilm (Fig. 3A; Fig. S1A through C). Similarly, Fig. 3B shows a markedly higher number of downregulated proteins in the biofilm across all three pressure conditions.

Notably, biofilm-overexpressed proteins were primarily associated with the Cluster of Orthologous Groups (COG) category for signal transduction mechanisms (T), with a slight decrease in abundance under HHP, while proteins with unknown functions (S) showed an increase at elevated pressure conditions (Fig. 4; Table S1). Conversely, proteins more prevalent in planktonic cells were predominantly linked to energy production and conversion (C), amino acid transport and metabolism (E), and again, unknown functions (S). This likely reflects a higher metabolic activity of planktonic cells compared with biofilm. Intriguingly, these proteins appeared to be less influenced by variations in pressure (Fig. 4; Table S1).

Further analysis revealed that proteins involved in flagellar assembly (DPRO_2385), adhesin secretion system (DPRO_0300), EPS production (DPRO_1637), chemotaxis (DPRO_1513, 1866, 0986), and the putative sensor kinase RcsC (DPRO_3650) were overexpressed in the biofilm (Fig. 3C). The upregulation of these proteins aligns with their essential roles in biofilm formation.

Conversely, planktonic cells exhibited an increased abundance of proteins associated with sulfur metabolism (Fig. 3C and D). A substantial proportion of proteins involved in the dissimilatory sulfate reduction pathway was upregulated. DsvC (DPRO_0194), Sat (DPRO_2419), and AprA (DPRO_2421) were approximately sevenfold more abundant in planktonic cells compared with the biofilm, while DsvA (DPRO_1445) and DsvB (DPRO_1446) exhibited an approximately eihtfold increase (Fig. 3D).

Additionally, portions of the oxidative phosphorylation (DPRO_1558-1564), citrate cycle (DPRO_1936-1939), and biosynthesis of several amino acids (DPRO_2838, 2839, 2841, 2842, 2843, 2844, 2846, 2847, 2850) were upregulated in planktonic cells. While the exact reason for the downregulation of these proteins in biofilms remains to be explored, it may be attributed to the lower metabolic activity of biofilm, reducing their demand for these pathways. To determine whether these proteomic differences were related to the presence of metal, identical experiments were conducted using epoxy surfaces at 0.1 MPa. The expression profiles of all proteins cited above were essentially unchanged between metal and epoxy incubations, with nearly all proteins mentioned above following the same biofilm–planktonic patterns (Table S2).

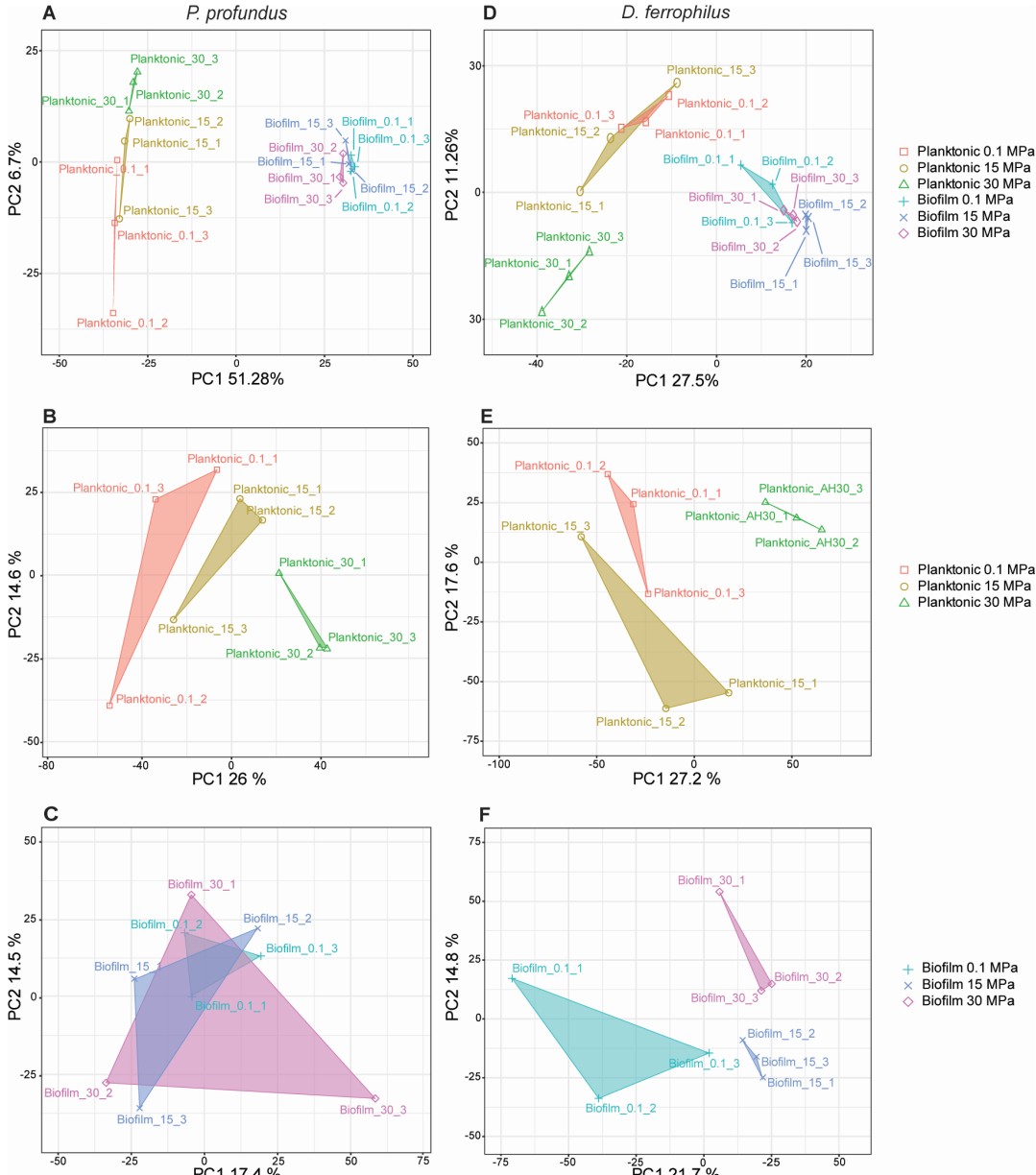

**FIG 2** Principal component analysis (PCA) plots of the two principal axis for proteins expression in *P. profundus* (A, B and C) and *D. ferrophilus* (D, E and F) cultures under all conditions (A and D) for planktonic cells (B and E) and biofilm (C and F). The areas of coverage were calculated by constructing convex hulls around the points of interest.

In contrast, *D. ferrophilus* proteomes showed a notably lower number of differentially expressed proteins, with 86, 105, and 204 proteins altered under biofilm or planktonic conditions at 0.1, 15, and 30 MPa, respectively. These proteins were relatively evenly distributed at 0.1 and 15 MPa (Fig. 5A and B; Fig. S1B); however, a sharp increase in those downregulated in biofilm was observed at 30 MPa (Fig. 5B; Fig. S1D).

The biofilm phase showed the upregulation of COG categories associated with signal transduction mechanisms (T) and energy production and conversion (C), peaking at 15 MPa. At 30 MPa, a significant decrease in chemotaxis proteins (COG category N) was observed. Finally, nearly all downregulated COG categories showed a positive correlation with pressure (Fig. 4).

To further explore pressure-induced proteomic changes, proteins identified as significantly modulated in biofilm or planktonic proteomes at each hydrostatic pressure

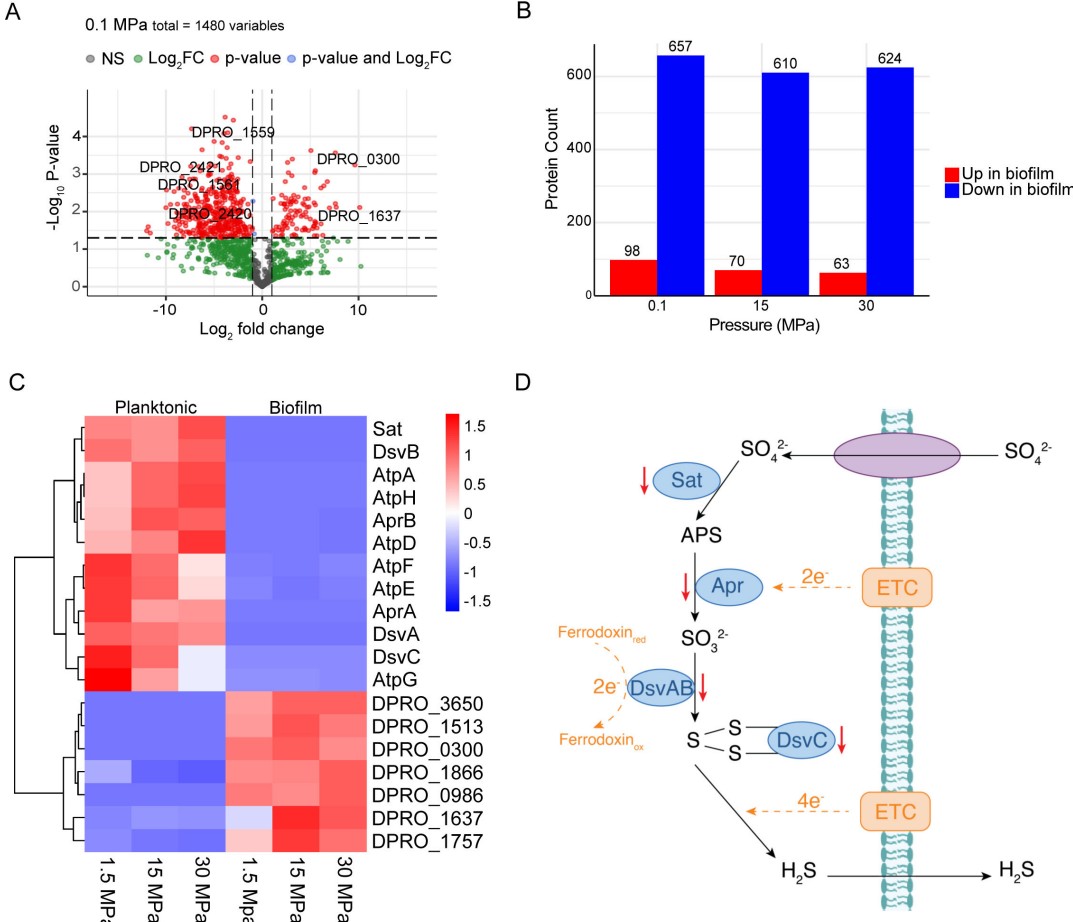

**FIG 3** Characterization of the functional relationship between planktonic cells and biofilms in *P. profundus*. (A) Representative volcano plot comparing differentially expressed proteins in biofilm over planktonic cells at 0.1 MPa culture. (B) A bar plot shows significantly altered proteins in biofilm over planktonic cells under three pressure conditions. (C) Heatmap indicating the proteins showing a significant differential expression in the biofilm proteome groups compared with the planktonic cells. (D) Dissimilatory sulfate reduction pathway at the biofilm was found to be impaired with core proteins Sat, Apr, DsvAB, and DsvC significantly downregulated.

were analyzed for their functional roles. Pathways related to metal binding, iron-sulfur cluster, transmembrane activities, signal transduction, and the CheY-like superfamily were predominantly enriched in the biofilm at 0.1 and 15 MPa, as shown by the balloon and Sankey plots (Fig. 5C through E). However, at 30 MPa, this trend was completely reversed, indicating that high hydrostatic pressure negatively impacts these molecular functions across diverse pathways, particularly those related to metal metabolism and binding (Fig. 5D and E).

Among the overexpressed proteins at 0.1 and 15 MPa, several were associated with key metabolic and biofilm-related pathways, DFE_2667 and DFE_0599 participate in pyruvate metabolism, while DEF_1385 (Idc) and DFE_0173 contribute to key steps of the TCA cycle. Additionally, 4Fe4S ferredoxins (DFE_2829 and DFE_0653), cytochrome c class III (DFE_1444), and cytochrome-c3 hydrogenase (DFE_1090) are involved in electron transfer and metal binding. Furthermore, c-di-GMP phosphodiesterases (DFE_3137) and metal-dependent c-di-GMP phosphodiesterases (DFE_3175) play roles in biofilm formation. The abundance of these proteins suggests that at low and moderate pressure, *D. ferrophilus* actively engages in redox processes, including enhanced electron transfer and metal binding, leading to its high corrosion capability. When the same experiments were performed using epoxy, these biofilm-enriched proteins, particularly those linked to electron transfer and metal binding, were no longer differentially expressed between

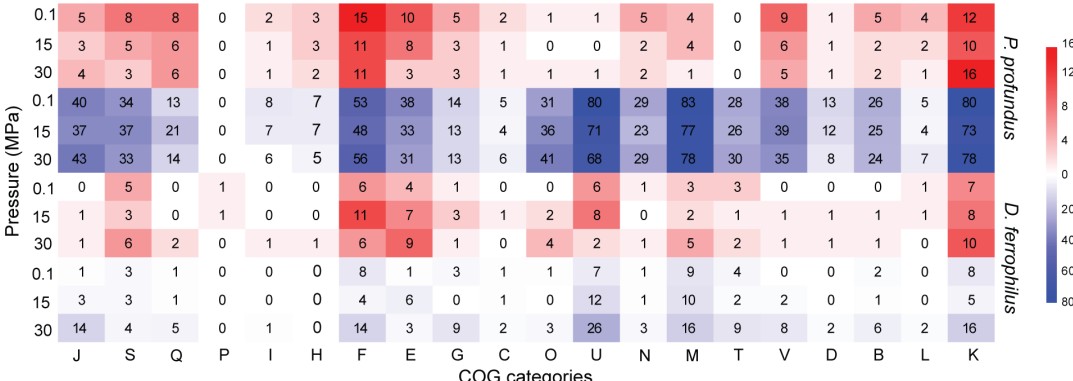

**FIG 4** Heat map showing the raw count of upregulated (red) and downregulated (blue) genes in biofilm samples at three different pressures for each species. Columns represent COG categories, while rows represent the two species under the indicated pressure conditions.

biofilm and planktonic cells, or were instead more abundant in planktonic cells (Table S2).

## Biofilm pressure-induced proteomic changes reveal species-specific correlation

To understand the direct impact of hydrostatic pressure on biofilm protein dynamics and, consequently, on the corrosive capabilities of both bacterial species in response to pressure changes, a comparative analysis of the biofilm proteomes at three pressure levels was conducted. Clustering analysis was employed to identify shared response patterns within the proteome, and proteins showing significant differential expression ($P$-value $\leq 0.05$) were detected. For both species, four main response categories could be defined as HHP inhibited, medium hydrostatic pressure (MHP) inhibited, MHP induced, and HHP induced (Fig. 6A, B and 7A through D; Tables S3 and S4).

Among the significantly differently expressed 49 proteins identified with the Fuzzy c-means clustering analysis and displaying differential expression in *P. profundus* biofilm, 14 were associated with the HHP inhibited category and primarily linked with signal transduction mechanisms (T), energy production (C), and lipid metabolism (I). The only six proteins inhibited at MHP were involved in amino acids transport and metabolism (E), transcription (K), replication, recombination, and repair (L) (Table 1; Table S3).

In contrast, the production of the remaining 29 proteins exhibited a positive correlation with increasing pressure. Notably, 13 proteins, predominantly involved in replication, recombination, and repair (L), energy production and conversion (C), and unknown (S), were more abundant at MHP (Table 1), while 16 proteins were overexpressed at 30 MPa, with functions related to translation (J), signal transduction mechanisms (T), cell wall (M), and amino acid and nucleotides transport and metabolism (E, F) (Table 1; Table S3).

MHP showed an increased abundance of proteins involved in energy metabolism, such as acetyltransferase (DPRO_2179) and NADH oxidoreductase (DPRO_0788) (Fig. 6C), while at HHP formate dehydrogenase (DPRO_2284), phosphoribosylpyrophosphate synthase (DPRO_3674) and metal-dependent phosphohydrolase (DPRO_2758), along with the aforementioned DPRO_2179 and DPRO_0788, were upregulated (Fig. 6C).

These findings clearly demonstrate that *P. profundus* exhibits enhanced metabolic and energy activity in the biofilm at higher pressure, contributing to the elevated corrosion rate.

The proteome of *D. ferrophilus* biofilms displayed more drastic changes in response to pressure, leading to the over- or underexpression of 65 proteins. The abundance of 31 proteins was increased at elevated pressures. Specifically, at MHP and HHP, the production of 15 and 16 proteins, respectively, was enhanced (Table S4).

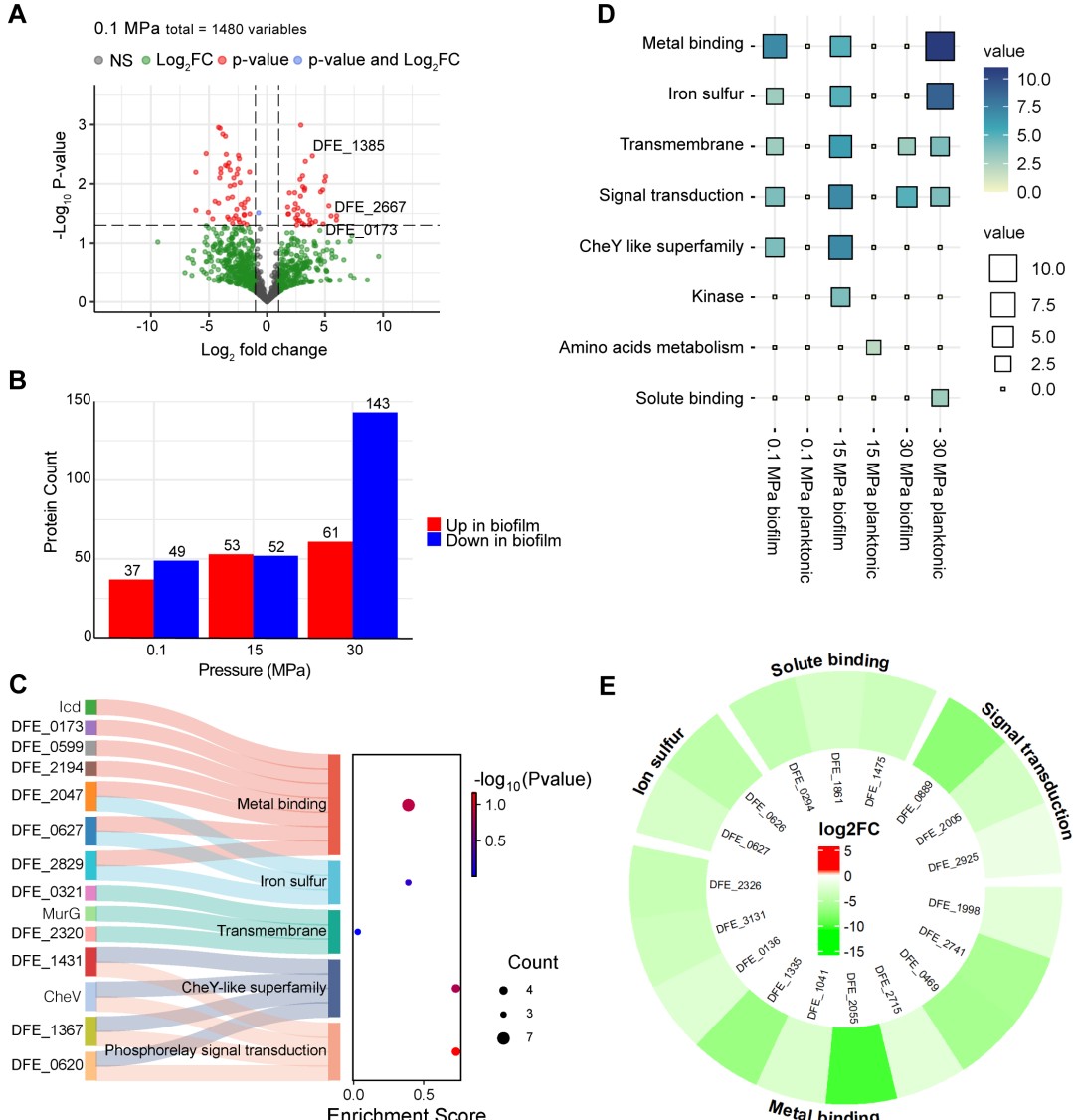

**FIG 5** In-depth proteomic characterization and functional categorization of altered proteins proteomes in *D. ferrophilus*. (A) Representative volcano plot comparing differentially expressed proteins in biofilm over planktonic cells at 0.1 MPa culture. (B) Average count of significantly altered proteins in biofilm over planktonic cells under three pressure conditions. (C) Balloon plot representing the functional characterization of proteins enriched in biofilm or planktonic cells under three pressure conditions. The values indicate the number of proteins identified in each pathway, represented by both color intensity and square size. (D) A Sankey plot displayed the enriched proteins and metabolic pathways they involved in biofilm under 0.1 MPa culture. (E) Circular heatmap indicating the downregulated proteins and their cellular function (outer part of the heatmap) in biofilm proteome compared with planktonic cells under 30 MPa.

At MHP, five proteins associated with cell wall/membrane/envelope biogenesis (M), including four involved in the biosynthesis of peptidoglycan (DFE_1818), GDP mannose (DFE_2001), glycan (DFE_2837), and glycolipids (DFE_2840) (Fig. 7E (i), Table S4) were upregulated in an attempt to adapt to the pressure changes. Meanwhile, none of the COG categories exhibited specific enrichment at the HHP group. Instead, proteins were evenly distributed among functional groups related to signal transduction mechanisms (T), energy production and conversion (C), molecule transport and metabolism (E, F, I) (Table 1).

Meanwhile, transcription (K) and amino acid transport and metabolism (E) were the predominantly represented categories among the 10 proteins downregulated under HHP. Similarly, at MHP, nine proteins involved in energy production and conversion (C)

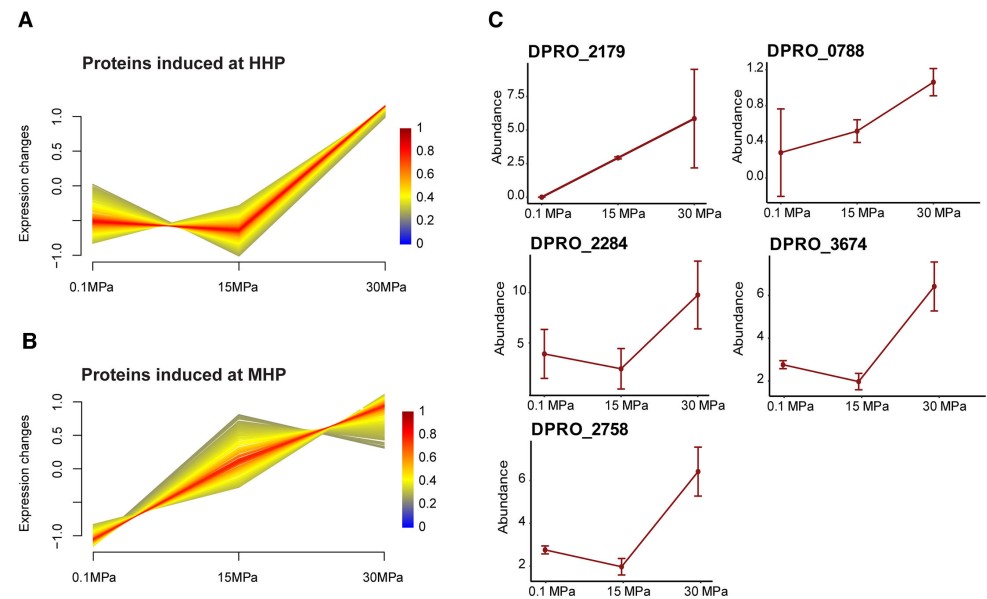

**FIG 6** Fuzzy c-means clustering analysis of biofilms shows the protein expression pattern and abundance of selected proteins associated with different pressures for *P. profundus*; colors represent the degree of membership (0–1) of each sample to the displayed cluster. Two patterns are displayed: proteins induced at HHP (A) and proteins induced at MHP (B). (C) Abundances of selected proteins enriched in each cluster were highlighted. Error bars indicate the calculated standard deviations of three biological replicates.

and amino acids transport and metabolism (E) collectively accounted for more than one-third of the 25 proteins underexpressed (Table 1).

Finally, the detrimental effect of hydrostatic pressure led to the downregulation of energy metabolism proteins DFE_2667 and DFE_2829, along with DFE_0001, DFE_1602, and DFE_0318. Additionally, amino acid production proteins DFE_0791 and DFE_3005 (Fig. 7E (ii-iii)-F) and transcription proteins DFE_2227, DFE_2389, and DFE_0378 (Fig. 7E (iv)-F) were also downregulated, despite being all previously found to be overexpressed at 0.1 MPa.

## DISCUSSION

MIC is a complex process shaped by the interplay of microbial physiology, microbial metabolism, and environmental conditions. This study aims to clarify how hydrostatic pressure influences the corrosion activity of two sulfate-reducing bacteria, *P. profundus* and *D. ferrophilus*, highlighting distinct adaptive strategies and metabolic shifts under different pressures.

Despite its inherently lower corrosive capacity, *P. profundus* exhibited an increasing corrosion rate with rising hydrostatic pressure. Although an abiotic control was not included in this study, the possibility that the observed corrosion was solely due to abiotic processes is unlikely. This assumption is supported by a recent study using the same medium and strain (45), which demonstrated that *P. profundus* causes significantly more aggressive corrosion than under abiotic conditions, with a comparable relative increase in corrosion rate between 0.1 and 30 MPa. In contrast, *D. ferrophilus* showed severe corrosion at 0.1 and 15 MPa, aligning with previous studies on other SRB species (52, 53). However, at 30 MPa, the corrosion rate dropped 10-fold, marking a significant decline in the bacterium's corrosive capability, likely due to its inability to maintain a high metabolic rate under extreme pressure. These species-specific responses highlight how diverse adaptive strategies to high-pressure conditions among bacteria with similar core metabolisms can directly influence their MIC potential. Understanding these physiologi-

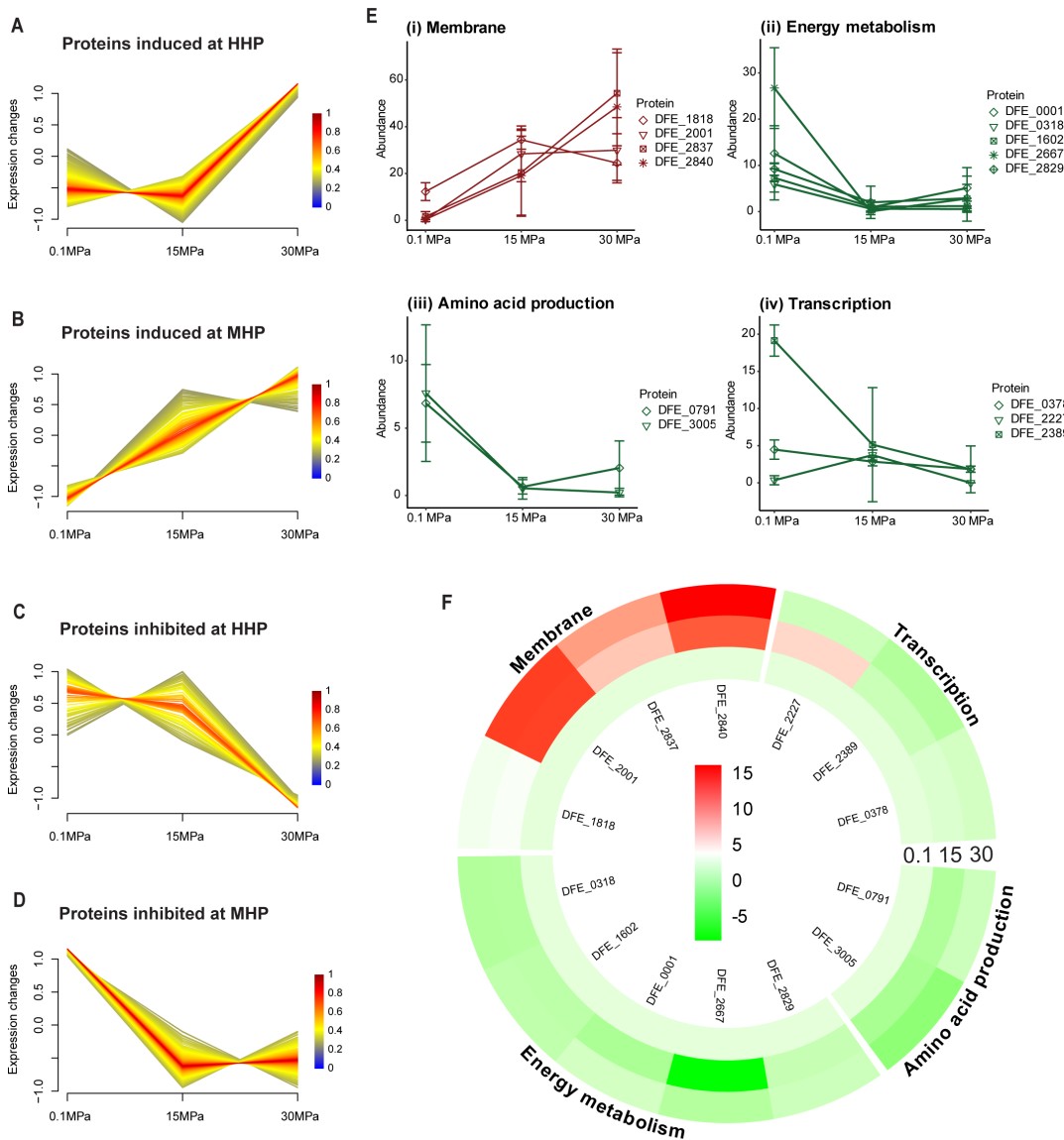

**FIG 7** Fuzzy c-means clustering analysis of biofilm shows the protein expression pattern associated with different pressures for *D. ferrophilus;* colors represent the degree of membership (0–1) of each sample to the displayed cluster. Four patterns, proteins induced at HHP (A), proteins induced at MHP (B), proteins inhibited at HHP (C), and proteins inhibited at MHP (D) were identified based on expression levels at three pressures (0.1, 15, and 30 MPa). (E) Abundances of selected proteins enriched in each cluster were highlighted with their functional annotations. Error bars indicate the calculated standard deviations of three biological replicates. (F) Circular heatmap indicating the log2FC of the principal proteins affected by hydrostatic pressure against 0.1 MPa. Protein symbols are indicated in the inner part of the heatmap and their cellular function in the outer part of the heatmap.

cal adaptations is essential for accurately predict corrosion risks to materials deployed in such environments.

The distinct responses observed here are in agreement with prior studies examining SRB adaptation to hydrostatic pressure. Pradel et al. (46) performed the first combined genomic and proteomic characterization of the deep-sea piezophile *Desulfovibrio piezophilus*, revealing pressure-responsive genes and proteins linked to amino acid transport, energy metabolism, and sulfate reduction. Several of these functions parallel the pressure-associated proteomic changes detected in our study, suggesting conserved metabolic strategies among piezophilic SRB. In contrast, the non-piezophilic *Desulfovibrio alaskensis* G20 was shown to exhibit reduced growth rates and enhanced biofilm formation under pressures up to 14 MPa (47). These findings support the notion that

**TABLE 1** Cluster of orthologous groups (COG) categories representing proteins whose expression was affected by changing in hydrostatic pressure in *P. profundus* and *D. ferrophilus* biofilms[a,b]

| COG categories | P. profundus | | | | D. ferrophilus | | | |
|---|---|---|---|---|---|---|---|---|
| | HHP inhibited | MHP inhibited | MHP induced | HHP induced | HHP inhibited | MHP inhibited | MHP induced | HHP induced |
| J - Translation, ribosomal structure, and biogenesis | 0 | 0 | 1 | 2 | 0 | 1 | 0 | 0 |
| K - Transcription | 1 | 1 | 0 | 0 | 3 | 1 | 0 | 0 |
| L - Replication, recombination, and repair | 1 | 1 | 2 | 0 | 0 | 1 | 1 | 0 |
| B - Chromatin structure and dynamics | 0 | 0 | 0 | 0 | 1 | 0 | 0 | 0 |
| D - Cell cycle control, cell division, chromosome partitioning | 0 | 0 | 0 | 1 | 0 | 0 | 0 | 0 |
| T - Signal transduction mechanisms | 6 | 1 | 1 | 2 | 0 | 1 | 2 | 2 |
| M - Cell wall/membrane/envelope biogenesis | 0 | 0 | 0 | 2 | 0 | 2 | 5 | 1 |
| N - Cell motility | 1 | 0 | 0 | 0 | 0 | 2 | 0 | 1 |
| U - Intracellular trafficking, secretion, and vesicular transport | 0 | 0 | 1 | 1 | 0 | 1 | 1 | 1 |
| O - Post-translational modification, protein turnover, chaperones | 0 | 1 | 0 | 0 | 0 | 2 | 2 | 0 |
| C - Energy production and conversion | 2 | 0 | 3 | 1 | 0 | 5 | 1 | 2 |
| G - Carbohydrate transport and metabolism | 0 | 1 | 0 | 0 | 0 | 0 | 0 | 0 |
| E - Amino acid transport and metabolism | 1 | 2 | 0 | 2 | 2 | 4 | 0 | 2 |
| F - Nucleotide transport and metabolism | 0 | 0 | 0 | 3 | 0 | 0 | 0 | 2 |
| H - Coenzyme transport and metabolism | 0 | 0 | 0 | 0 | 0 | 2 | 0 | 1 |
| I - Lipid transport and metabolism | 2 | 0 | 0 | 0 | 1 | 0 | 0 | 2 |
| P - Inorganic ion transport and metabolism | 0 | 0 | 0 | 0 | 0 | 0 | 0 | 1 |
| Q - Secondary metabolites biosynthesis, transport, and catabolism | 0 | 0 | 1 | 1 | 1 | 0 | 0 | 0 |
| S - Function unknown | 2 | 1 | 3 | 1 | 2 | 2 | 2 | 2 |

[a]The discrepancy between the sum of counts and the number of differently expressed proteins was due to the dual functionality of certain proteins or the presence of proteins falling into unknown COG categories.
[b]HHP and MHP inhibited/induced refer to proteins which expression decreased or increased at 30 or 15 MPa, respectively.

pressure not only modulates SRB metabolism but also reshapes biofilm physiology, with potential implications for MIC.

MIC is indeed closely linked to biofilm formation, and investigations into the biochemical framework of SRB biofilms have consistently shown distinct protein expression profiles compared to their planktonic counterparts (54, 55). To explore pressure-induced adaptive strategies, shotgun proteomics was performed on both biofilm and planktonic cells of the two species, aiming to identify key proteins and metabolic pathways involved in these responses. The PCA plots showed a clear separation between planktonic and biofilm samples, with each forming distinct clusters, consistent with previous findings (54–57). Notably, the biofilm samples exhibited much tighter clustering with a smaller distribution area compared to the more dispersed planktonic samples across both species. This suggests that the proteome composition in biofilms is more conserved, potentially reflecting a more uniform physiological state under biofilm-associated conditions.

The randomized PCA distribution across different conditions in *P. profundus* suggested a low sensitivity to pressure fluctuations, a characteristic trait of non-strict piezophilic bacteria. This aligns with its physiology, which exhibits higher metabolism and cell division at ~10–15 MPa (42), while maintaining broad tolerance to both higher or lower pressures (45).

Moreover, *P. profundus* exhibited downregulation of the entire dissimilatory sulfate reduction pathway under biofilm conditions, indicating a lower metabolic state likely resulting from limited availability of an electron donor. This reduced activity, along with the absence of outer membrane cytochromes, likely contributes to its limited corrosion capabilities. As MIC is closely associated with active biofilm formation, the inability of biofilm-associated *P. profundus* cells to harness electrons released from abiotic metal dissolution, a key mechanism of electric MIC (EMIC) (38), further underscores its restricted corrosion potential. Instead, this bacterium appears to rely on the oxidation of organic carbon sources, such as L-lactate (equations 1 to 3), which are more readily accessible to planktonic cells.

$$CH_3CHOHCOO^- + NAD^+ \rightarrow CH_3COCOO^- + NADH + H^+ \tag{1}$$

$$CH_3COCOO^- + CoA\text{-}SH + 2Fd_{ox} \rightarrow CH_3CO\text{-}SCoA + CO_2 + 2Fd_{red} \tag{2}$$

$$CH_3CO\text{-}SCoA + ADP + P_i \rightarrow CH_3COO^- + ATP + CoA\text{-}SH \tag{3}$$

This pathway provides four electrons for the reduction of sulfate to sulfide (equations 4 to 6):

$$SO_4^{2-} + ATP \rightarrow APS + PP_i \tag{4}$$

$$APS + 2e^- + H^+ \rightarrow SO_3^{2-} + AMP \tag{5}$$

$$SO_3^{2-} + 6e^- + 6H^+ \rightarrow H_2S + 3H_2O \tag{6}$$

Overall reaction:

$$2CH_3CHOHCOO^- + SO_4^{2-} + 2H^+ \rightarrow 2CH_3COO^- + H_2S + 2CO_2 + 2H_2O \tag{7}$$

This specific behavior exemplifies the less aggressive chemical MIC (CMIC), where corrosion primarily results from biogenic $H_2S$ production (equation 7) (39, 40), which is further diminished when the $H_2S$ reacting with dissolved iron, precipitates in form of iron sulfide (FeS) deposits which creates a protective layer (45).

This hypothesis was further supported by the epoxy control incubations, in which proteomic expression patterns were virtually identical to those on steel, confirming that the presence of metal does not influence *P. profundus* physiology, and consistent with its apparent incapacity for extracellular electron uptake.

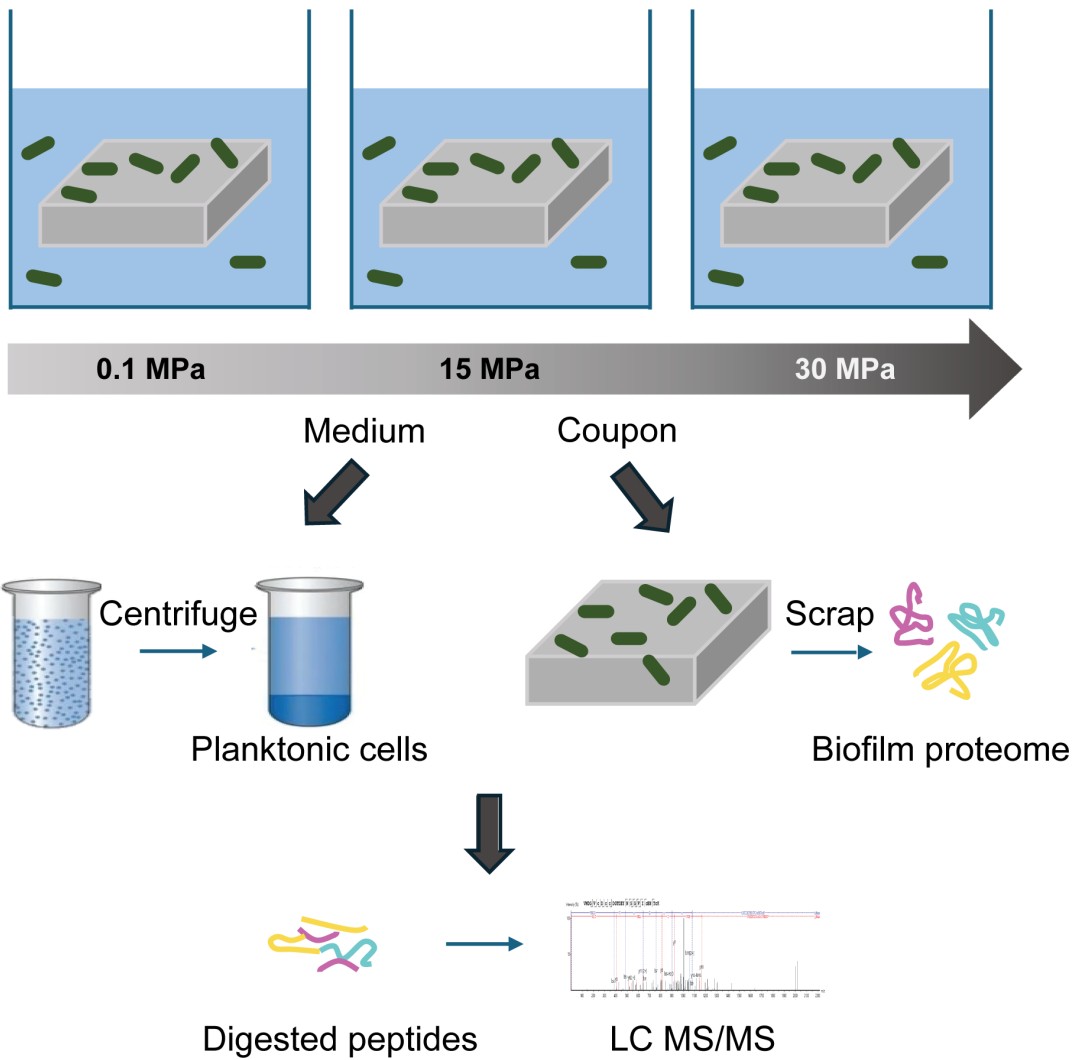

**FIG 8** Study design for proteomic analysis of bacteria under three pressure conditions. (i) Culture media were collected and centrifuged to isolate planktonic cells. (ii) Biofilms were recovered from the coupons by scraping. (iii) Proteins from both planktonic cells and biofilms were extracted, digested, and subjected to LC-MS/MS analysis.

Our experimental data confirmed the influence of hydrostatic pressure on *P. profundus* corrosive ability through the analysis of the biofilm proteome changes across three pressure conditions. While overall proteins production remained largely unaffected, a slight rise in biofilm metabolic activity—evidenced by increased energy metabolism proteins—was observed at MHP, corresponding to a minor rise in corrosion rate. At HHP, a further metabolic increase, coupled with reduced cells attachment and a weakened biofilm-protective effect (45), contributed to an additional increment in the corrosion rate.

Notably, numerous differentially expressed proteins with unknown functions were detected across all conditions. Among these, only two were consistently present at all pressures, linking them directly to the biofilm state. Three proteins were exclusive to 15 and 30 MPa, suggesting a role in pressure adaptation, while seven were found only at 0.1 and 30 MPa, indicating a broader involvement in stress response. While further molecular experiments are required for validation, these findings offer new insights into potential candidates involved in the pressure adaptation of piezophilic bacteria.

Unlike *P. profundus*, *D. ferrophilus* exhibited strong pressure-related clustering in PCA analyses of both planktonic and biofilm proteomes, corresponding to a pronounced

physiological response to hydrostatic pressure. At 0.1 and 15 MPa, its biofilm phase displayed high metabolic activity, driven by enriched energy metabolism. Functional pathway analysis revealed an increased abundance of iron-binding proteins predicted to be periplasmic or secreted and aligning with *D. ferrophilus* ability to harness electrons released during corrosion (equation 8) to drive sulfate reduction, thereby facilitating highly aggressive EMIC (equation 9) (49, 51):

$$Fe^0 \rightarrow Fe^{2+} + 2e^- \tag{8}$$
$$Fe^0 + SO_4^{2-} + 8H^+ \rightarrow Fe^{2+} + H_sS + 4H_2O \tag{9}$$

On the contrary, in the epoxy control, these enrichments were absent or markedly reduced, and in few cases, the same proteins were more abundant in planktonic cells. This shift further indicates that the presence of steel directly stimulates specific metabolic pathways in response to the availability of electrons.

However, at 30 MPa, proteomic data suggested a metabolic shift favoring a planktonic lifestyle, likely leading to the loss of EMIC activity resulting in a corrosion rate comparable to *P. profundus* ($P > 0.05$) and consistent with the less aggressive CMIC.

Biofilm proteome analysis across increasing pressures also suggested membrane modifications at MHP, a common adaptive response to environmental stressors (58).

Overall, the inability of *D. ferrophilus* to adapt to HHP was evident in the negative correlation between pressure and the production of energy metabolism and amino acid biosynthesis proteins, leading to a strong metabolic inhibition. While some key proteins were already underexpressed at 15 MPa, a significant decline in the corrosion rate was only observed at 30 MPa. This discrepancy may be attributed to the critical role of numerous pressure-affected proteins with unknown functions, as well as a reduced growth rate.

In conclusion, this study uncovers distinct hydrostatic pressure-related stress responses employed by two SRB species, with CMIC representing the main mechanism under this specific extreme condition. These findings emphasize that MIC critically depends on bacteria ability to form biofilms on metal surfaces while maintaining high metabolic activity. Moreover, our results highlight species-specific corrosion strategies, emphasizing that environmental conditions and adaptive responses to stress play a critical role in shaping MIC processes. However, many pressure-affected proteins remain uncharacterized, underscoring the need for further research into extremophile metabolism. A comprehensive approach, considering not only target species but also their physiological responses to extreme conditions, is essential for enhancing the prediction and assessment of corrosion risks in the deep sea and other extreme environments.

## MATERIALS AND METHODS

### Bacterial cultivation

Both *P. profundus* 500-1 and *D. ferrophilus* IS5 were purchased from the German Collection of Microorganisms and Cell Cultures GmbH (DSMZ, Germany). Initial cultures were performed in anaerobic *Desulfobacter* sp. 195C medium (DSMZ-German Collection of Microorganisms and Cell Cultures GmbH, Germany) from glycerol stocks for 48 h at atmospheric pressure (0.1 MPa). Subsequently, 40 mL of fresh 195C medium was added to each culture, and the pressure was adjusted to either 0.1, 15, or 30 MPa. Once reached the exponential phase, the medium was discarded after centrifugation at 10,000 × *g* for 8 min at 4°C, and the cells were resuspended in anaerobic artificial seawater (ASW) medium 1 (National Center of Marine Algae and Microbiota, USA) (composition: 471 mM NaCl, 56.5 mM MgCl$_2$, 27.5 mM MgSO$_4$·7H$_2$O, 9.7 mM KCl, 6.8 mM CaCl$_2$·2H$_2$O, 287 µM K$_2$HPO$_4$, 10 mM NaHCO$_3$ and 0.1% of Mineral and Wolfe's vitamins solution) amended with 100 ppm L-cysteine, 2.2 mM ,L-lactate and 40 mM MOPS and with pH adjusted to 7.8, upon reaching an OD$_{600}$ equivalent to 0.02.

For this study, three biological replicates per condition were performed. Each replicate consisted of 50 mL of either *P. profundus* or *D. ferrophilus* culture together with an AH36 mild steel coupon, sealed in a sterile aluminum bag and pressurized to final hydrostatic pressure for 7 days. Epoxy coupons, prepared and incubated under identical conditions but limited to 0.1 MPa, were included as a control to assess the influence of metal on microbial physiology.

## Material preparation and corrosion rate

AH36 mild steel metal coupons were used to test the corrosion rate and as substrate for biofilm formation.

Each coupon measured 10 × 10 × 2 mm and was sequentially polished to a final grit US800. After acetone washing and ultrapure water rinsing, each coupon was UV sterilized for 30 min on each side before incubation.

Corrosion rate was calculated through weight loss analysis, out of the three biological replicates, following ASTM G1-03 protocol (59), which consists of weighing the coupons at the beginning of the incubation and again at the end after the elimination of biofilm and corrosion product.

Successively, the measured weight loss is applied to equation 10:

$$\text{Corrosion rate} = (K \times W)/(A \times T \times D) \tag{10}$$

where $K$ is constant ($0.876 \times 10^4$), $W$ is weight loss (g), $T$ is duration of the incubation (h), $A$ is area ($cm^2$) of the specimen, and $D$ is density of the steel ($g/cm^3$), which allows for the extrapolation of the rate of corrosion in mm/year.

## Biofilm proteins scraping

Coupons for proteomics analysis were snap frozen in liquid nitrogen immediately after incubation and preserved at −80°C. For protein extraction, each coupon was thawed and transferred into a sterile plastic container containing 1.5 mL Tris-urea buffer (8 M urea, 50 mM Tris, pH 8). Biofilm was removed by thoroughly scraping the entire coupon surface with a sterile biofilm scraper (Fig. 8). Each scraper was used for a single coupon to prevent cross-contamination.

The biofilm suspension in Tris-urea buffer, with the coupon remaining in the solution, was sonicated using a Vibra-Cell probe sonicator VXC 750 (Sonic & Materials, Inc., USA) at 20 kHz with a power output of 15 W with 3 bursts of 45 s. Following sonication, the entire suspension was transferred to an Eppendorf tube and centrifuged at 20,000 × *g* for 10 min. The supernatants were carefully transferred to fresh tubes for subsequent steps.

## Planktonic proteins extraction

Planktonic cells were centrifuged at 10,000 × *g* for 8 min at 4°C immediately after the end of the incubation (Fig. 8). The supernatant was discharged, and the cells frozen, preserved, and treated following the same procedure adopted for the biofilm, with the exception of using 100 µL of Tris-urea buffer instead of 1.5 mL.

## In solution tryptic digestion

Proteins quantification was performed using Pierce BCA Protein Assay Kits (Thermo Fisher Scientific, USA). The protein samples were reduced with 10 mM Tris (2-carbox-yethyl) phosphine (TCEP) for 20 min at 25°C and alkylated with 55 mM chloroacetamide (CAA) for 30 min at room temperature in the dark. Sample mixture was diluted with a ratio of 1:7 with 100 mM triethylammonium bicarbonate buffer (TEAB), thereby reducing the urea concentration to less than 1 M. This was followed by first digestion at 25°C with lys-C at 1:20 enzyme to protein (wt/wt) ratio for 2 h and second digestion at 25°C with trypsin at a 1:40 trypsin to protein ratio overnight (18 h). Tryptic-digested peptides were desalted using HLB PRiME 96-well C18 plates (Waters Corporation, USA). Elution was

performed with 250 µL of 65% acetonitrile (ACN), 0.1% formic acid (FA) buffer. Peptide quantification was performed using Pierce Quantitative Peptide Assays & Standards (Thermo Fisher Scientific, USA). The eluted peptides were then dried using a vacuum concentrator, and the desired amount was reconstituted for injection with 3% ACN, 0.1% acetic acid, and 0.06% trifluoroacetic acid (TFA).

## Label-free mass spectrometry data acquisition

After rehydration to a final concentration of 1 µg/µL, the peptides were analyzed through LC-MS/MS using an EASY-nLC 1200 System coupled to Orbitrap Fusion Lumos Tribrid Mass Spectrometer (Thermo Fisher Scientific, USA). A flow rate of 300 nL/min using an EASY-Spray 75 µm × 50 cm column packed with PepMap C18 3 µm, 100 Å (Thermo Fisher Scientific, USA) was applied for peptides mixture separation, at a controlled temperature of 50°C. A 70-min gradient liquid chromatography was performed using mobile phase A, consisting of 0.1% formic acid (FA) in HPLC water, followed by multiple steps using a mobile phase B consisting of 0.1% FA in 95% ACN. The steps were: 45 min 2%–27%, 15 min 27%–50%, 5 min 50%–90%, and 5 min 90%. After separation, the peptides were analyzed using an electron-spray potential of 2.1 kV. A full MS scan (350–1,550 m/z range) was run at a resolution of 60,000 and a maximum injection time of 100 ms. The resolution of the higher collision-induced dissociation (CID) spectra was applied for fragmentation. For the full MS scan and the MS2 scan, the automatic gain control (AGC) was set at 100% and 150%, respectively. For the identification of peptides and post-translational modifications, the mass spectrometer was utilized in positive mode, employing data-dependent acquisition of MS2. Exclusion criteria for MS/MS included single and unassigned charged ions.

## Data analysis

Proteome Discoverer software (Version 2.5, Thermo Fisher Scientific, San Jose, CA, https://proteomesoftware.com/) was used for database search. *P. profundus* (Taxon ID 57320) and *D. ferrophilus* (Taxon ID 241368) database were downloaded from UniProt. Raw LC–MS/MS results were searched against the database using the Sequest HT search engine. The precursor tolerance was set as 20 ppm and fragment ion tolerance at 0.8 Da. Carbamidomethylation of cysteine was specified as static modification. Oxidation of methionine, deamidation of asparagine and glutamine were set as dynamic modifications. The results were exported to text file for further analysis.

R statistical software (v4.3.3) (60) was employed for all statistical analyses. The expression of the proteins was analyzed for biofilm against planktonic cells at each hydrostatic pressure. A two-tailed Student's *t*-test ($P$-value ≤ 0.05) was applied to identify the differently expressed proteins which were further filtered (fold change < −2 or > 2) and classified against EggNOG 5.0 database (61) to identify their functional category, KEGG GENES database (62) to investigate their specific functions and ProtCompB 9.0 (63).

The functional analysis *P. profundus* was primarily conducted manually due to the limited availability of mapped metabolic pathways. The functional categorization and pathway analysis for *D. ferrophilus* was performed using the Database for Annotation, Visualization, and Integrated Discovery (DAVID) (64)

Successively, the abundance of the proteins produced in the biofilm was normalized against the total count and clustered using the Fuzzy c-means (FCM) with R package Mfuzz (65).

A one-way ANOVA ($P$-value ≤ 0.05) was employed to identify proteins that exhibited significant differential expression, within each cluster, by comparing the mean abundances of each individual protein under all pressures.

The differently expressed proteins were again classified against EggNOG 5.0 (61) and KEGG GENES databases (62).

## ACKNOWLEDGMENTS

This study was supported by the National Research Foundation, Prime's Minister's Office, Singapore, under its Competitive Research Program (Award: NRF-CRP21-2018-0102). Additional support was provided by the Singapore's Center for Environmental Life Sciences Engineering of Nanyang Technological University, Singapore. R.M.S. was supported by A*STAR Core funding, Singapore.

## AUTHOR AFFILIATIONS

[1]Singapore Centre for Environmental Life Sciences Engineering, Nanyang Technological University, Singapore, Singapore

[2]Institute of Molecular and Cell Biology (IMCB), Agency for Science, Technology and Research (A*STAR), Singapore, Singapore

[3]Luminis Water Technologies, Singapore, Singapore

## AUTHOR ORCIDs

Nicolò Ivanovich http://orcid.org/0000-0002-5432-3716
Radoslaw M. Sobota http://orcid.org/0000-0002-2455-2526
Federico M. Lauro http://orcid.org/0000-0002-8373-1014

## FUNDING

| Funder | Grant(s) | Author(s) |
|---|---|---|
| National Research Foundation Singapore | CRP21-2018-0102 | Nicolò Ivanovich |

## AUTHOR CONTRIBUTIONS

Nicolò Ivanovich, Conceptualization, Formal analysis, Investigation, Writing – original draft, Writing – review and editing | Xue Guo, Formal analysis, Investigation, Writing – original draft, Writing – review and editing | Radoslaw M. Sobota, Resources, Funding acquisition, Writing – review and editing | Federico M. Lauro, Conceptualization, Project administration, Resources, Funding acquisition, Writing – review and editing

## DATA AVAILABILITY

The raw proteomic spectra and search data were uploaded to the Jpost repository with the following accession number: JPST004092.

## ADDITIONAL FILES

The following material is available online.

### Supplemental Material

**Supplemental figure and tables (Spectrum01590-25-s0001.docx).** Figure S1; Tables S1 to S4.

### Open Peer Review

**PEER REVIEW HISTORY (review-history.pdf).** An accounting of the reviewer comments and feedback.

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
