## [Reviewer comments · Microbiology Spectrum]

Microbiology Spectrum

ADAPTATION TO HYDROSTATIC PRESSURE MODULATES PROTEOME DYNAMICS IN CORROSIVE SULFATE- REDUCING BACTERIA

Nicolò Ivanovich, Xue Guo, Radoslaw Sobota, and Federico Lauro

Corresponding Author(s): Nicolò Ivanovich, Nanyang Technological University

Review Timeline:

Submission Date:	May 22, 2025
Editorial Decision:	July 2, 2025
Revision Received:	August 20, 2025
Accepted:	September 5, 2025

Editor: Katharina Kujala

Reviewer(s): Disclosure of reviewer identity is with reference to reviewer comments included in decision letter(s). The following individuals involved in review of your submission have agreed to reveal their identity: Satoshi Wakai (Reviewer #1)

Transaction Report:

DOI: <https://doi.org/10.1128/spectrum.01590-25>

Re: Spectrum01590-25 (ADAPTATION TO HYDROSTATIC PRESSURE SHAPES PROTEOME DYNAMICS IN CORROSIVE SULFATE-REDUCING BACTERIA)

Dear Dr. Nicolò Ivanovich:

Thank you for the privilege of reviewing your work. Below you will find my comments, instructions from the Spectrum editorial office, and the reviewer comments.

Revision Guidelines

Sincerely,
Katharina Kujala
Editor
Microbiology Spectrum

Reviewer #1 (Comments for the Author):

This manuscript focuses on microbially influenced corrosion in deep-sea environments, with particular attention to the effects of pressure, and provides a detailed examination of both planktonic and sessile microorganisms within corrosion products. The topic is highly interesting; however, there are several issues that prevent a comprehensive evaluation of the relationship between pressure, corrosion, and microbial metabolism.

One major limitation is the absence of abiotic control in the corrosion experiments, which makes it difficult to determine whether the observed corrosion is truly metabolism-driven. Additionally, the study does not assess sulfate-reducing activity under each pressure condition, leaving uncertainty about whether the cells were metabolically active or in a dormant state. The term "biofilm" is used throughout the manuscript, yet what is being observed appears to be cells embedded within corrosion products rather than structured communities on material surfaces. Thus, it is questionable whether this qualifies as a biofilm in the conventional sense, and I recommend using more careful language to describe these observations.

Ideally, a non-corrosive substrate such as titanium or polymer could be employed to promote biofilm formation independently of corrosion processes. This would allow for a clearer comparison of the physiological characteristics of planktonic cells, surface-attached biofilm-forming cells, and cells embedded within corrosion products. While I do not request such additional experiments in this study, it is important to acknowledge that, under the current setup, it is difficult to disentangle the contributions of surface attachment, biofilm development, and corrosion-associated processes to microbial activity. A clearer experimental distinction between these conditions would greatly enhance the interpretability of the results.

Moreover, the test medium used for corrosion experiments contains 2.2 mM lactate, which must be considered. Planktonic cells would have utilized lactate as their primary energy source, while sessile cells within corrosion products may have been exposed to both lactate and metallic iron as electron donors. This context is crucial when attempting to link pressure, metabolism, and corrosion processes. Overall, a more careful and nuanced interpretation is required for the conclusions drawn in this manuscript.

Specific Comments:

Figure 1: There is no abiotic control data included. As a result, it is unclear whether the pressure-dependent variation in corrosion rate observed with *P. profundus* is due to microbial activity or simply the physical effects of pressure. Particularly at 30 MPa, the corrosion rate is comparable to that of *D. ferrophilus*, but no statistical comparison has been made against the background. Additionally, discontinuous axes are generally inappropriate in scientific figures as they may distort visual interpretation. Please consider revising the graph accordingly.

Figure 2: While pressure-dependent variation is observed in all cases except the biofilm data of *P. profundus*, the data for biofilm and planktonic cells are plotted in separate graphs for each microbial species. As a result, direct comparisons between these two populations cannot be made based on the current figure layout. Please either refrain from drawing conclusions about their relationship or redesign the figure to allow a clearer visual comparison between biofilm and planktonic states for each strain.

References:

The following references should be cited and discussed in relation to the effects of pressure on gene expression in sulfate-reducing bacteria:

<https://doi.org/10.1371/journal.pone.0055130>

<https://doi.org/10.3389/fmicb.2018.01465>

Additionally, more recent publications on corrosion in deep-sea environments are available and may enhance the manuscript's context:

<https://doi.org/10.3389/fmicb.2025.1540664>

<https://doi.org/10.1264/jsme2.me23089>

<https://doi.org/10.1016/j.jmst.2024.07.026>

Reviewer #2 (Comments for the Author):

Ivanovich and colleagues aimed to investigate the metabolic pathways involved in microbially influenced corrosion (MIC) at high pressure, as in deep sea. They studied the proteome composition of two bacterial species, *Pseudodesulfovibrio profundus* and *Desulfovibrio ferrophilus*, as known to be involved in the microbially influenced corrosion (MIC). In particular, *P. profundus* is a piezophilic bacterium able to colonize environment with high pressure, while *D. ferrophilus* in this study serves as control. The bacteria were cultivated, in presence of a metal coupons, a 0.1 MPa, 15 MPa and 30 MPa. The metal corrosion was also evaluated.

The manuscript is generally well written, and I can suggest only some limited corrections.

Here all my suggestions:

In the entire manuscript, the thousands are expressed without any separator, they should be expressed using a comma as a separator (1000 -> 1,000), even in case on number lists (1,500, 1,200, ...).

I found Figure S1 very interesting. Considering the importance of biofilm for the MIC process, I think the authors should consider including this figure (biofilm vs free cells PCA) in the main text, for instance as a panel of Figure 2. I also suggest to the authors to discuss the fact that the areas relative to biofilm is quite lower than that of planktonic, both for the two species. This could suggest that the proteome composition is more conserved in biofilm.

The legend of Figure 4 should explain more in detail the figure. For example it is not clear to me the use of the colors. Furthermore, I suggest coloring the heatmap on the basis of standardized evaluations of the pathway enrichment, for instance using the residuals of chi square. Indeed, the number of genes of a certain pathway can also depend on the total number of genes of that pathway in the organism (it varies among pathways and organisms).

The legend Figure 5 should explain more in deep the figure. For instance, it is not clear to me what is the "value" in panel C. Check the bold of the letters in the legend: the letter relative to panel C must be bolded.

The legend of Figure 6 should does not explain the colors used in panel A and B

The legend of Figure 7 should does not explain the colors used in panel A-D

Line 359, do you refer to biofilm bacteria or biofilm and planktonic?

The legend of Figure 8 should explain better what is reported in the figure

Reply to Reviewer#1

We sincerely thank the reviewer for their constructive comments. We believe our revision directly respond to the concerns raised and substantively enhance the manuscript's clarity, coherence, and scientific rigor. Detailed responses to each comment are provided below. Reviewer's comments are reproduced in blue, with our responses in black beneath each comment. Line numbers refer to the "Marked-Up Manuscript" file, where all modifications have been highlighted in yellow.

"This manuscript focuses on microbially influenced corrosion in deep-sea environments, with particular attention to the effects of pressure, and provides a detailed examination of both planktonic and sessile microorganisms within corrosion products. The topic is highly interesting; however, there are several issues that prevent a comprehensive evaluation of the relationship between pressure, corrosion, and microbial metabolism."

Author's reply: We thank the reviewer for the thoughtful assessment of our manuscript and for recognizing the relevance of our study on microbially influenced corrosion (MIC) in deep-sea environments.

"One major limitation is the absence of abiotic control in the corrosion experiments, which makes it difficult to determine whether the observed corrosion is truly metabolism-driven."

Author's reply: We agree with the reviewer that including an abiotic control could provide additional clarity to our results, particularly in confirming that the observed corrosion in *P. profundus* cultures is not exclusively due to abiotic processes. However, the primary objective of our study was to investigate the effect of pressure changes on the proteomic profiles of the analysed strains and to assess their impact on corrosion activity.

We consider it unlikely that the corrosion measured in *P. profundus* cultures was solely the result of abiotic processes. This is supported by a recent study using the same medium and strain (doi: 10.3389/fmicb.2025.1540664), which demonstrated that *P. profundus* induces significantly more aggressive corrosion than the abiotic control. Although the corrosion rates reported in that study were higher due to longer incubation times and greater medium availability, the relative increase in corrosion rate between 0.1 and 30 MPa is highly comparable (58% in our study vs. 56% in the cited work). We therefore consider this published study as robust background knowledge supporting our conclusions.

We have now included a brief explanation, to provide context and support for our interpretation of the results (lines : 297-302): "Although an abiotic control was not included in this study, the possibility that the observed corrosion was solely due to abiotic processes can be considered unlikely, as supported by a recent study using the same medium and strain (48), which demonstrated that *P. profundus* causes

significantly more aggressive corrosion than under abiotic conditions, with a comparable relative increase in corrosion rate between 0.1 and 30 MPa.”

“Additionally, the study does not assess sulfate-reducing activity under each pressure condition, leaving uncertainty about whether the cells were metabolically active or in a dormant state. The term "biofilm" is used throughout the manuscript, yet what is being observed appears to be cells embedded within corrosion products rather than structured communities on material surfaces. Thus, it is questionable whether this qualifies as a biofilm in the conventional sense, and I recommend using more careful language to describe these observations.”

Author’s reply: We agree with the reviewer that assessing sulfate-reducing activity under each pressure condition could provide valuable insights into the metabolic state of the cells. Unfortunately, such measurements were not performed in this study, and the available samples do not allow for retrospective determination of sulfate-reducing activity. While direct evidence of sulfate reduction is not available, the observed differences in corrosion rates and proteomic profiles across pressure conditions suggest that the cells remained metabolically active and responsive to environmental changes.

The reviewer’s concern regarding the use of the term “biofilm” is acknowledged. However, according to the widely accepted definition by Flemming and Wingender (2010), a biofilm is “a microbial community embedded in a self-produced matrix of hydrated extracellular polymeric substances (EPS) that provides mechanical stability, mediates adhesion to surfaces, and forms a cohesive, three-dimensional polymer network interconnecting and immobilizing the cells”. In corrosion-associated systems, EPS production and matrix formation often occur concomitantly with the precipitation of corrosion products, which may integrate into the biofilm structure.

Such composite structures are well documented in microbially influenced corrosion field and fall within the scope of the biofilm definition provided by Flemming and Wingender. Therefore, we believe the use of the term “biofilm” in this study is consistent with established scientific usage.

“Ideally, a non-corrosive substrate such as titanium or polymer could be employed to promote biofilm formation independently of corrosion processes. This would allow for a clearer comparison of the physiological characteristics of planktonic cells, surface-attached biofilm-forming cells, and cells embedded within corrosion products. While I do not request such additional experiments in this study, it is important to acknowledge that, under the current setup, it is difficult to disentangle the contributions of surface attachment, biofilm development, and corrosion-associated processes to microbial activity. A clearer experimental distinction between these conditions would greatly enhance the interpretability of the results.”

Author's reply: We thank the reviewer for this valuable suggestion.

The experiments were indeed replicated using epoxy coupons; however, the results were initially considered only as an internal control.

We have now incorporated a direct comparison between the selected proteins of interest for both *P. profundus* and *D. ferrophilus* at 0.1 MPa into the results section and in Table S2.

Lines 215 198-202: "To determine whether these proteomic differences were related to the presence of metal, identical experiments were conducted using epoxy surfaces at 0.1 MPa. The expression profiles of all proteins cited above were essentially unchanged between metal and epoxy incubations, with nearly all proteins mentioned above following the same biofilm–planktonic patterns (Table S2)."

Lines 232-235: "When the same experiments were performed using epoxy, these biofilm-enriched proteins, particularly those linked to electron transfer and metal binding, were no longer differentially expressed between biofilm and planktonic cells, or were instead more abundant in planktonic cells (Table S2)."

We have also integrated the information to the discussion section:

Lines 366-369: "This hypothesis was further supported by the epoxy control incubations, in which proteomic expression patterns were virtually identical to those on steel, confirming that the presence of metal does not influence *P. profundus* physiology, consistent with its apparent incapacity for extracellular electron uptake."

Lines 394-397: "On the contrary, in the epoxy control, these enrichments were absent or markedly reduced, and in few cases, the same proteins were more abundant in planktonic cells. This shift further indicates that the presence of steel directly stimulates specific metabolic pathways related to the availability of electrons."

Finally we have added the following to the methods section:

Lines 440-442: "Epoxy coupons, prepared and incubated under identical conditions but limited to 0.1 MPa, were included as a control to assess the influence of metal on microbial physiology."

The raw data will also be added to the repository cited in the data availability.

"Moreover, the test medium used for corrosion experiments contains 2.2 mM lactate, which must be considered. Planktonic cells would have utilized lactate as their primary energy source, while sessile cells within corrosion products may have been exposed to both lactate and metallic iron as electron donors. This context is crucial when attempting to link pressure, metabolism, and corrosion processes. Overall, a more careful and nuanced interpretation is required for the conclusions drawn in this manuscript."

Author's reply: We thank the reviewer for this comment and for emphasizing the importance of considering the test medium.

The ability of SRB to utilise lactate as an electron donor is well established in the literature. In the discussion, the focus was placed on biofilm activity to explain the

impact of pressure on corrosion rates. The different strategies adopted by the two species were addressed in detail, with the hypothesis that *P. profundus* is unable to utilise electrons released during the corrosion process and relies solely on lactate as an electron donor. This would result in chemical MIC (CMIC) and correspondingly lower corrosion rates (lines 342-365). In contrast, the increased metabolic activity observed in *D. ferrophilus* biofilms was interpreted as the result of their ability to utilise electrons released from the corroding surface, leading to electrical MIC (EMIC) and higher corrosion rates (lines 384-393).

Specific Comments:

“Figure 1: There is no abiotic control data included. As a result, it is unclear whether the pressure-dependent variation in corrosion rate observed with *P. profundus* is due to microbial activity or simply the physical effects of pressure. Particularly at 30 MPa, the corrosion rate is comparable to that of *D. ferrophilus*, but no statistical comparison has been made against the background. Additionally, discontinuous axes are generally inappropriate in scientific figures as they may distort visual interpretation. Please consider revising the graph accordingly.”

Author’s reply: We appreciate reviewer’s comment.

Please refer to our previous response regarding the absence of an abiotic control. With respect to the plot, the original intention was to enable a clear visual comparison between the two species. However, we understand the reviewer’s concern and have revised the figure to a multi-facet layout with independent y-axes, allowing for improved readability while maintaining the comparability of trends.

“Figure 2: While pressure-dependent variation is observed in all cases except the biofilm data of *P. profundus*, the data for biofilm and planktonic cells are plotted in separate graphs for each microbial species. As a result, direct comparisons between these two populations cannot be made based on the current figure layout. Please either refrain from drawing conclusions about their relationship or redesign the figure to allow a clearer visual comparison between biofilm and planktonic states for each strain.”

Author’s reply: To enhance the clarity of both the results and the discussion, the PCA of all samples has been relocated from the supplementary material to Fig. 2. We believe this revised layout allows the distinction between proteomic profiles to be more clearly compared across both species and conditions.

References:

The following references should be cited and discussed in relation to the effects of pressure on gene expression in sulfate-reducing bacteria:

<https://doi.org/10.1371/journal.pone.0055130>

<https://doi.org/10.3389/fmicb.2018.01465>”

Author's reply: We have added a paragraph to integrate the suggested papers in our discussion (line 311-321): “The distinct responses observed here are in line with prior studies examining SRB adaptation to hydrostatic pressure. N. Pradel et al. (45) performed the first combined genomic and proteomic characterization of the deep-sea piezophile *Desulfovibrio piezophilus*, revealing pressure-responsive genes and proteins linked to amino acid transport, energy metabolism, and sulfate reduction. Several of these functions parallel the pressure-associated proteomic changes detected in our study, suggesting conserved metabolic strategies among piezophilic SRB. In contrast, the non-piezophilic *Desulfovibrio alaskensis* G20 was shown to exhibit reduced growth rates and enhanced biofilm formation under pressures up to 14 MPa (46). These findings support our observation that pressure not only modulates SRB metabolism but also reshapes biofilm physiology, with potential implications for MIC.”

“Additionally, more recent publications on corrosion in deep-sea environments are available and may enhance the manuscript's context:
<https://doi.org/10.3389/fmicb.2025.1540664>”

Author's reply: We thank the reviewer for the appreciation of our work and for highlighting this paper. The paper you mentioned was published by our group, with myself as the first author, and it has already been cited in the manuscript.

“<https://doi.org/10.1264/jsme2.me23089>
<https://doi.org/10.1016/j.jmst.2024.07.026>”

Author's reply: The suggested citations have been added to the introduction (lines 75-77, line 101).

References:

Flemming, HC., Wingender, J. The biofilm matrix. *Nat Rev Microbiol* **8**, 623–633 (2010). <https://doi.org/10.1038/nrmicro2415>

Reply to Reviewer#2

We sincerely thank the reviewer for their insightful comments. We greatly appreciate reviewer's effort in carefully evaluating our work, which has helped us to substantially improve the clarity, coherence, and overall quality of the manuscript. Detailed responses to each comment are provided below. Reviewer's comments are reproduced in blue, with our responses in black beneath each comment. Line numbers refer to the "Marked-Up Manuscript" file, where all modifications have been highlighted in yellow.

"Ivanovich and colleagues aimed to investigate the metabolic pathways involved in microbially influenced corrosion (MIC) at high pressure, as in deep sea. They studied the proteome composition of two bacterial species, *Pseudodesulfovibrio profundus* and *Desulfovibrio ferrophilus*, as known to be involved in the microbially influenced corrosion (MIC). In particular, *P. profundus* is a piezophilic bacterium able to colonize environment with high pressure, while *D. ferrophilus* in this study serves as control. The bacteria were cultivated, in presence of a metal coupons, a 0.1 MPa, 15 MPa and 30 MPa. The metal corrosion was also evaluated."

Author's reply: We thank the reviewer for the clear and accurate summary of our study's aims and experimental design. We appreciate the recognition of the relevance of investigating metabolic pathways involved in microbially influenced corrosion (MIC) under high-pressure conditions, and the rationale behind selecting *Pseudodesulfovibrio profundus* as a piezophilic model and *Desulfovibrio ferrophilus* as a comparative control.

The manuscript is generally well written, and I can suggest only some limited corrections.

Here all my suggestions:

"In the entire manuscript, the thousands are expressed without any separator, they should be expressed using a comma as a separator (1000 -> 1,000), even in case on number lists (1,500, 1,200, ...)."

Author's reply: We appreciate the effort put by the reviewer in ensuring a consistent formatting of numbers in the manuscript. We have revised the formatting to include commas as thousand separators throughout the text. Specifically, we have corrected the expressions at line 109 ("1500" to "1,500" and "3000" to "3,000"), line 145 ("3168" to "3,168" and "2842" to "2,842"), along with a thorough check to ensure consistency across the entire manuscript.

"I found Figure S1 very interesting. Considering the importance of biofilm for the MIC process, I think the authors should consider including this figure (biofilm vs free cells PCA) in the main text, for instance as a panel of Figure 2. I also suggest to the

authors to discuss the fact that the areas relative to biofilm is quite lower than that of planktonic, both for the two species. This could suggest that the proteome composition is more conserved in biofilm.”

Author’s reply: We appreciate reviewer’s insightful suggestions. We have reorganized Figure 2 and combined it with the original Figure S1. The revised Figure 2 now more clearly illustrates the distinct proteomic patterns between planktonic cells and biofilm groups in both SRB bacteria.

We greatly appreciate your thoughtful perspective, which prompted a deeper interpretation of our data. In response, we have incorporated the following additions to the revised Discussion section (lines 328-334 380): “The PCA plots show a clear separation between planktonic and biofilm samples, with each forming distinct clusters, consistent with previous findings (51-54). Notably, the biofilm samples exhibit much tighter clustering with a smaller distribution area compared to the more dispersed planktonic samples across both species. This suggests that the proteome composition in biofilms is more conserved, potentially reflecting a more uniform physiological state under biofilm-associated conditions.”

“The legend of Figure 4 should explain more in detail the figure. For example it is not clear to me the use of the colors. Furthermore, I suggest coloring the heatmap on the basis of standardized evaluations of the pathway enrichment, for instance using the residuals of chi square. Indeed, the number of genes of a certain pathway can also depend on the total number of genes of that pathway in the organism (it varies among pathways and organisms).”

Author’s reply: We thank the reviewer for their constructive suggestion. We agree that visualizing differential expression data using standardized values from pathway enrichment analyses (e.g., chi-square residuals) can account for differences in the total number of genes per pathway between organisms. However, such an approach requires a comprehensive pathway annotation, which is currently limited for *P. profundus* in publicly available databases such as KEGG. As a result, a robust and comparable enrichment analysis could not be performed for this species.

To ensure consistency between the two organisms, we therefore chose to present the data as a heat map based on the raw up/down counts of genes within each COG category, without normalization to pathway size. We have clarified the figure legend to better explain the colour scale and structure of the plot (lines: 564-567): “Heat map showing the raw count of upregulated (red) and downregulated (blue) genes in biofilm samples at three different pressures for each species. Rows represent COG categories, while columns represent the two species under the indicated pressure conditions.”

“The legend Figure 5 should explain more in deep the figure. For instance, it is not clear to me what is the "value" in panel C. Check the bold of the letters in the legend: the letter relative to panel C must be bolded.”

Author’s reply: We thank the reviewer for pointing out the confusing parts and helping us improve the clarity of our manuscript. In response, we have revised the figure caption to provide a more explicit explanation. The values represent the number of proteins identified in each group and are indicated by both the darker colour and the size of the squares.

The revised figure caption (line 573-576) is as follows: “**C**. Balloon plot representing the functional characterization of proteins enriched in biofilm or planktonic cells under three pressure conditions. The values indicate the number of proteins identified in each pathway, represented by both colour intensity and square size.”

The legend of Figure 6 should does not explain the colors used in panel A and B

Author’s reply: We acknowledge the missing explanation in Figure 6. We have updated the figure legend and caption to include a clear description of the colours used in panels A and B (lines 583-584): “colours represent the degree of membership (0–1) of each sample to the displayed cluster”.

The legend of Figure 7 should does not explain the colors used in panel A-D

Author’s reply: Similar to the previous question, we have added the colours to the legend (lines 589-590).

Line 359, do you refer to biofilm bacteria or biofilm and planktonic?

Author’s reply: We thank the reviewer for the question. Our statement refers primarily to the biofilm-associated cells of *P. profundus*. The ability to harness electrons released from abiotic metal dissolution typically requires direct contact with the metal surface, which is characteristic of biofilm cells rather than planktonic cells. Therefore, the observed limited corrosion potential is mainly attributed to the biofilm bacteria’s inability to perform this electron uptake, while planktonic cells are unlikely to contribute significantly to this process.

In order to avoid misinterpretation the sentence has been adjusted (lines 346-349): “As MIC is closely associated with active biofilm formation, the inability of biofilm-associate *P. profundus* cells to harness electrons released from abiotic metal dissolution, a key mechanism of electric MIC (EMIC) (37), further underscores its restricted corrosion potential”.

The legend of Figure 8 should explain better what is reported in the figure

Author's reply: We agree with the reviewer on this point. We have updated the figure caption accordingly, and the revised version (lines 599-604) is also provided below: "**Figure 8.** Study design for proteomic analysis of bacteria under three pressure conditions. (1) Culture media were collected and centrifuged to isolate planktonic cells, (2) Biofilms were recovered from the coupons by scraping. (3) Proteins from both planktonic cells and biofilms were extracted, digested, and subjected to LC-MS/MS analysis.

Re: Spectrum01590-25R1 (ADAPTATION TO HYDROSTATIC PRESSURE MODULATES PROTEOME DYNAMICS IN CORROSIVE SULFATE-REDUCING BACTERIA)

Dear Dr. Nicolò Ivanovich:

Your manuscript has been accepted, and I am forwarding it to the ASM production staff for publication. Your paper will first be checked to make sure all elements meet the technical requirements. ASM staff will contact you if anything needs to be revised before copyediting and production can begin. Otherwise, you will be notified when your proofs are ready to be viewed.

Sincerely,
Katharina Kujala
Editor
Microbiology Spectrum